# African Gene Flow Reduces Beta-Ionone Anosmia/Hyposmia Prevalence in Admixed Malagasy Populations

**DOI:** 10.3390/brainsci11111405

**Published:** 2021-10-25

**Authors:** Harilanto Razafindrazaka, Veronica Pereda-Loth, Camille Ferdenzi, Margit Heiske, Omar Alva, Minah Randriamialisoa, Caroline Costedoat, Michel Signoli, Thierry Talou, Monique Courtade-Saidi, Anne Boland, Jean-François Deleuze, Catherine Rouby, Chantal Radimilahy, Thierry Letellier, Moustafa Bensafi, Denis Pierron

**Affiliations:** 1Aix Marseille University, CNRS, EFS, ADES, 13344 Marseille, France; caroline.costedoat@univ-amu.fr (C.C.); michel.signoli@univ-amu.fr (M.S.); 2Institute for Advanced Study in Toulouse, 21 Allées de Brienne, CEDEX 6, 31015 Toulouse, France; 3Equipe de Médecine Evolutive, URU EVOLSAN, Faculté de Chirurgie Dentaire, Université de Toulouse III, 31073 Toulouse, France; veronica.pereda@univ-tlse3.fr (V.P.-L.); margitheiske@gmail.com (M.H.); omalsa01@gmail.com (O.A.); thierry.letellier@inserm.fr (T.L.); 4Groupement Scientifique en Biologie et Médecine Spatiale, Faculté Médecine Rangueil, 31400 Toulouse, France; 5CNRS, UMR5292, Lyon Neuroscience Research Center, University Lyon, 69000 Lyon, France; camille.ferdenzi-lemaitre@cnrs.fr (C.F.); cathy.rouby@gmail.com (C.R.); moustafa.bensafi@cnrs.fr (M.B.); 6Département Histoire, Faculté des Lettres et des Sciences Humaines, Université d’Antananarivo, Antananarivo 101, Madagascar; mirandriami@gmail.com; 7Laboratoire de Chimie Agro-Industrielle, Université de Toulouse, INP-ENSIACET, 31030 Toulouse, France; Thierry.Talou@ensiacet.fr; 8Laboratoire d’Histologie-Embryologie, Faculté de Médecine Rangueil, 31062 Toulouse, France; monique.courtade-saidi@univ-tlse3.fr; 9Centre National de Recherche en Génomique Humaine (CNRGH), Institut de Biologie François Jacob, CEA, 91057 Evry, France; boland@cng.fr (A.B.); deleuze@cng.fr (J.-F.D.); 10Institut de Civilisations/Musée d’Art et d’Archéologie, Université d’Antananarivo, Isoraka, Antananarivo 101, Madagascar; radimilahych@gmail.com

**Keywords:** admixture, beta-ionone, olfaction, specific anosmia, Madagascar

## Abstract

While recent advances in genetics make it possible to follow the genetic exchanges between populations and their phenotypic consequences, the impact of the genetic exchanges on the sensory perception of populations has yet to be explored. From this perspective, the present study investigated the consequences of African gene flow on odor perception in a Malagasy population with a predominantly East Asian genetic background. To this end, we combined psychophysical tests with genotype data of 235 individuals who were asked to smell the odorant molecule beta-ionone (βI). Results showed that in this population the ancestry of the OR5A1 gene significantly influences the ability to detect βI. At the individual level, African ancestry significantly protects against specific anosmia/hyposmia due to the higher frequency of the functional gene (OR ratios = 14, CI: 1.8–110, *p*-value = 0.012). At the population level, African introgression decreased the prevalence of specific anosmia/hyposmia to this odorous compound. Taken together, these findings validate the conjecture that in addition to cultural exchanges, genetic transfer may also influence the sensory perception of the population in contact.

## 1. Introduction

Olfactory perception varies considerably from one individual to another, to such an extent that the same odorous stimulus can trigger various hedonic reactions, ranging from disgust to pleasure [1,2]. This olfactory diversity may be associated with changes in choices and behaviors toward specific food products [1]. A characteristic example of this olfactory diversity is so-called specific anosmia, in which certain specific molecules (such as androstenone or β-Ionone, βI) are well detected by one part of the human population, while the other part has little or no sensitivity [3]. Specific anosmia are sometimes associated with molecules produced by the human body and considered by some to be pheromones involved in inter-personal communication/human mate choice [4] (e.g., androstenone). However, specific anosmia can also impact the detection of molecules that are not significantly produced by the human body but are widely present in the human diet, such as beta-ionone. Therefore, specific anosmia may influence behavior toward certain foods, and it has been suggested that the percentage of anosmic in a population may influence cultural eating practices [5].

Education, age, sex, context, experience, as well as the physical and cultural environment play an important role in the development of inter-individual diversity of perception [6,7,8,9,10]. Nevertheless, the role of genetic background is increasingly proposed as a factor influencing odor perception and its hedonic component [11,12,13,14,15]. In particular, βI specific anosmia is one of the best examples of genetically driven perception. βI is a key aroma in numerous plants and fruits such as tomatoes, raspberries, and beverages such as some orange juices and wines [10,16,17]. Jaeger et al. (2013) showed that this molecule activates the olfactory receptor OR5A1 and that individuals carrying the functional form of OR5A1 are able to detect a 100-fold lower concentration of βI than individuals possessing only the non-functional form of OR5A1. The variant involved (rs6591536: G>A) substitutes an aspartic acid for asparagine at position 183 (N183D) in the second extracellular loop of the receptor, thereby decreasing its ability to be activated by βI. Interestingly, Jaeger and colleagues also showed that the OR5A1 genotype influences individuals’ behavior toward foods, beverages, and even personal and household care products containing βI [14,18].

Therefore, genetic differences between populations could also induce differences in odor perception at the population level. Notably, the functional form of the OR5A1 gene is the minor allele in Asia, whereas it is the most common allele in Africa [19]. Theoretically, this suggests that gene flow between these continents (i.e., through migration) could influence the level of specific anosmia and subsequently food preference in a resulting mixed population such as the Malagasy population resulting from an admixture of African and Asian populations [20]. This possibility also means that, in the context of cultural globalization, gene flow could influence the perception and preferences of a population receiving gene flow from another continent. Human migrations must therefore be considered when studying the history of a population’s food and scent preferences and related cultural practices.

However, to test whether gene flow can influence population odor perception, two important prerequisites must be demonstrated. (i) The first is to show that variants such as rs6591536 have a similar impact on individual βI-perception across different human populations and especially in admixed populations with significant biological and cultural differences from the populations where the genotype/phenotype relationship has been identified. The association between rs6591536 and βI-anosmia/hyposmia has only been published by Jaeger et al. (2013), and it is well known that only a limited number of genetic associations can be generalized and replicated in other human populations [13,21]. Indeed, GWAS (Genome-Wide Association Study) frequently produces false-positive results [22] and, more importantly, differences in genetic background can also influence the phenotype [13,21]. This is particularly important given the high genetic diversity of olfactory receptors that exist in human populations [23,24,25]. Genetic and epigenetic factors could impede OR5A1 expression and function in different populations. Alternatively, some populations might possess other functional olfactory receptors capable of detecting βI. It is therefore necessary to test the effect of rs6591536 on admixed populations (such as Malagasy populations). (ii) If the first prerequisite is confirmed, the second prerequisite is to demonstrate that in an admixed population, βI perception is indeed impacted by the continental origin of the olfactory receptor locus and thus by intercontinental gene flow. Indeed, even if βI perception is influenced by the allelic frequency of rs6591536, this rs6591536 allelic frequency in an admixed population can only marginally reflect the admixture rate. Indeed, several other factors not related to gene flow come into play. First, random fluctuation of allelic frequencies occurs over generations (genetic drift). Second, allelic frequencies can be influenced by selective constraints directly on the gene under study or on a neighboring locus (influencing the frequency of the allele considered by hitchhiking). It is therefore necessary to test the effect of the continental origin of the OR5A1 locus on βI perception.

This paper proposes to assess these two prerequisites in a mixed population living in the central highlands of the island of Madagascar that shows a primary ancestry of East Asian origin (>65%). Archaeological, ethnographic, and genetic data converge on the same observation that the colonization of the central highlands of Madagascar was a recent event by a population primarily of Indonesian origin but with great cultural and genetic enrichment from African populations [8,9]. Specifically, the objective of the present research is to (1) test the hypothesis that the strong genotype/phenotype association between rs6591536/βI found by Jaeger et al. (2013) is confirmed in an Asian/African admixed population despite genetic and cultural differences; (2) examine the impact of intercontinental gene flow on phenotypic diversity in the highland population, and more specifically, whether African gene flow may have impacted βI susceptibility in this population.

## 2. Materials and Methods

### 2.1. Participants

University students originating from Antananarivo participated in the study. Inclusion criteria were: (i) at least three of their four grandparents originating from the Antananarivo area; (ii) age between 18 and 33 years old; (iii) smoking fewer than five cigarettes a day; (iv) no subjects to allergies and nasal polyposis. Participants were also asked to attend the testing sessions without wearing any perfume, aftershave, face cream, or deodorant, and to fast for at least one hour prior to the test. On the basis of kinship coefficients estimated by King software [26], one case of cryptic relatedness at the 3rd degree was identified among two couple participants, in consequence, two individuals randomly chosen were excluded (it has to be noted that the inclusion or exclusion of this one individual does not change the results). A total of 235 participants responded to all of these criteria including 154 men and 83 women with a median age of 23 years (SD = 2.82). All participants gave their written informed consent for the olfactory testing procedures and for donating their DNA. The study was conducted according to the Declaration of Helsinki and approved by the Ethical Committee of the Ministry of Health of Madagascar.

### 2.2. Material

All experimental procedures were performed with 15 mL standardized glass vials (1.7 cm diameter at opening; 5.8 cm high), containing a scentless polypropylene fabric (3 × 7 cm; 3M, Valley, NE, USA) to optimize evaporation and air/oil partitioning, embedded either with a solution of mineral oil mixed with a specific concentration of βI, or only with mineral oil. All tests were performed in a regular classroom at the University of Antananarivo. Food grade βI (CAS: 14901-07-6) was obtained from Sigma-Aldrich France^®^ (St. Quentin Fallavier, France). The language used in the written documents (instructions, questionnaires) was French, but participants were allowed to answer in French or Malagasy.

### 2.3. Analysis

The study was conducted in two steps: perceptual evaluations and detection thresholds (step 1) and genotyping (step 2).

Step 1: Perceptual evaluations and detection threshold

The odor perception test was performed by asking all participants (*n* = 235) to sniff a vial containing a 20,000 ppm solution of βI for three seconds. Then, they were asked to rate the odor using the following 7-point scales: Hedonic (1: extremely unpleasant, 2: very unpleasant, 3: unpleasant, 4: neutral, 5: pleasant, 6: very pleasant, 7: extremely pleasant); Intensity (1: absent (no odor), 2: very low, 3: low, 4: medium strong, 5: strong, 6: very strong, 7: extremely strong); Familiarity (1: not at all familiar, 2: very little familiar, 3: little familiar, 4: medium familiar, 5: familiar, 6: very familiar, 7: extremely familiar); Edibility (would you eat it? 1: absolutely not, 2: certainly not, 3: probably not, 4: maybe, 5: probably, 6: certainly, 7: absolutely). Finally, participants were asked to freely identify and/or describe the odor.

One month later, an odor detection test was performed on 206 participants among 235 include initially using a sequence of six 4-alternative forced-choice tests, following the procedure described in Rouby et al. (2011) [27]. The smelling kit consisted of six rows of four glass vials. In each row, only one vial contained βI at a specific concentration, while the three others were blanks (no odor). Each row had a tenfold increasing concentration of βI from 2.05 × 10^−7^ mol/L to 2.05 × 10^−2^ mol/L (0.02 ppm to 2000 ppm). The participant’s task was to sniff each of the four vials in a row for three seconds with both nostrils, and to state which vial differed from the other three vials (i.e., which vial contained the odor). They evaluated whether they perceived nothing, a slight odor, or a strong odor in the designated vial. Each concentration was presented only once, and concentrations were presented in ascending order. We identified the individual detection threshold for βI as being the lowest concentration of βI that the individual identified correctly as well as all subsequent highest concentrations. This method is based on the principle that an individual who is able to detect the odor at a given concentration will not make any mistakes at higher concentrations. Since 25% of the individuals might choose the last correct vial by chance only, we used the concentration threshold of 2.05 × 10^−3^ to discriminate between sensitive and insensitive individuals.

Step 2: Genotyping

After detection tests, participants gave a saliva sample for genotyping. rs691536 was genotyped for all individuals using Illumina HumanOmni2.5-8 (Omni2.5) BeadChip and analyses were performed using PLINK 1.9 and R. KING software used to check family relationship by estimating kinship coefficients (https://people.virginia.edu/~wc9c/KING/manual.html, accessed on 1 July 2021).

### 2.4. Olfactory Phenotype and Perception Specificity

#### 2.4.1. Olfactory Phenotype

According to Mazzenta et al. [28], threshold data was clustered per age (5 years) in order to define the olfactory phenotype using a nonparametric Kruskal–Wallis test, α = 0.05 (16–20 N = 85; 21–25 N = 99; 26–33 N = 22). This olfactory phenotype should reflect the progression of the absolute olfactory threshold through the human life span [28].

#### 2.4.2. Testing the Perception Specificity

Participants were asked to smell a second odorant molecule tetrahydrothiophene (gas odor, 1127, 22.5%) a very common odorant molecule in the participants’ environment. The task consists of an identification performance based on a 4-choice test (gas, vinegar, plastic, coffee). As stated by Rouby et al. “*These familiar and thus ‘ecological’ odorants were used instead of the classical l-butanol because the latter also stimulates the trigeminal nerve*” [27]. The odorant molecules were trapped in sealed microcapsules (aminoplast type, diameter: 4–8 micro). The microcapsule-based ink was printed on cardboard paper (SILK-250g; Dimension: 11 cm × 21 cm). The odorant was printed on a delimited area (2 cm^2^ disc). The release of the odorant is done simply by rubbing the reserve of printed microcapsules [29].

### 2.5. Genome Data Analysis

#### 2.5.1. GWAS Analysis (Genome-Wide Association Study)

We perform a GWAS analysis for ßI perception in the Malagasy cohort following the recommendation presented in Marees, et al. [30]. Based on a dataset file comprising 2314914 SNPs for 206 individuals, we performed a quality-control procedure, removing sites with missing proportions >0.2 and a Hardy–Weinberg equilibrium filter at 1 × 10^−6^. We continued through the following steps using 2,262,562 sites. Using Plink 1.07 [31], we performed association analyses and visualization was done using R (package “qqman”).

From the HORDE database (https://genome.weizmann.ac.il/horde/, accessed on 1 July 2021 [32]), we listed all SNPs deletions and insertions altering the amino-acid sequence of olfactory receptors known to be activated by the beta-ionone molecule in in vitro studies (i.e., OR5A2, OR4D6, OR4D9, OR2B3, OR52D1, OR1G1) [13,30,31]. We specifically looked at the presence of these SNPs in our dataset and test their association with beta-ionone perception. We also compared the frequency of these SNPs across 8 Asian and African populations.

#### 2.5.2. Estimating Local Ancestry: Inference of Ancestry for Each Allele

Local ancestry assignment across the genome was performed based on HumanOmni2.5-8 data using the ELAI algorithm [33] and unphased data from the 1000 Genomes Project [27], with a pool of all African individuals as the [26] source and a pool of all Asian individuals as the Asian source. Succinctly, ELAI uses a two-layer hidden Markov model to detect the structure of haplotypes allowing to model two scales of linkage disequilibrium (one within a group of haplotypes and one between groups), and thereby infer local ancestry of admixed individuals. Based on the genotyping dataset of the Illumina HumanOmni2.5-8 covering the whole chromosome 11, this allows identifying the ancestral origin of a distinct chromosomal segment within each individual genome. The local estimates targeted the rs691536 locus of the OR5A1 gene localized on chromosome 11.

## 3. Results

### 3.1. Genotyping

Among 235 participants from the highlands of Madagascar, genotyping of SNP rs6591536 revealed 97 individuals homozygous with an AA genotype, 105 individuals heterozygous with an AG genotype, and 33 individuals homozygous GG (Appendix A). The A allele is predominant with an allele frequency of 63.6% and there is no significant deviation from the Hardy–Weinberg equilibrium (Fisher’s exact test *p*-value > 0.5). This result suggests that the sample is from an unstructured population, which is adequate for the proposed study.

### 3.2. Olfactory Phenotype and Perception Specificity

We found no difference in terms of the beta-ionone threshold of perception per age-class (Kruskal–Wallis test: *p* > 0.223).

The general anosmia test performed on a tetrahydrothiophene molecule identification (gas) did not reveal any significant difference between the genotype groups (group rs6591536 AA, AG GG, Fisher exact test *p* > 0.327). All groups had a very low level of anosmia to the tetrahydrothiophene. Of the 206 participants selected, 96.61% (N = 199) recognized the smell of gas. Of the seven participants unable to recognize the odor, three were AA (on rs6591536) and the remaining four carried the G allele. We did not exclude any individuals based on this analysis since the 3% who did not identify the smell of gas might be specifically anosmic or hyposmic to tetrahydrothiophene.

### 3.3. β-Ionone Perception according to Genotype

To examine the association between βI sensitivity and rs6591536, we first measured participants’ perceptual ratings and verbal description of the odorant (supraliminal concentration) before assessing their detection threshold with six 4-alternative forced-choice tests, following the procedure described in Rouby et al. [27].

#### 3.3.1. Perceptual Evaluations

At the supraliminal concentration (2 × 10^−1^ mol/L, 20,000 ppm), Malagasy participants (*N* = 235) evaluated βI as pleasant but only as moderately intense and familiar (Figure 1A, see Methods). According to their genotype the values obtained are (mean ± SD): Hedonic: AG 4.58 ± 1.35, GG 5.00 ± 1.30, AA 4.18 ± 1.14; intensity: AG 4.14 ± 1.12, GG 4.36 ± 0.78, AA 3.22 ± 1.06; familiarity: AG 3.73 ± 1.55, GG 4.12 ± 1.52, AA 2.84 ± 1.58; edibility: AG 3.79 ± 1.52, GG 4.00 ± 1.52, AA 3.30 ± 1.49 (Appendix A).

Analyses of variance (ANOVAs) with genotype as a between-subject factor showed that there were significant perceptual differences between AA, AG, and GG carriers (Hedonic: F 2.232 = 5.992, *p*-value = 0.002; intensity: F 2.232 = 25.01, *p*-value < 0.0001; familiarity: F 2.232 = 12.34, *p*-value < 0.001; edibility: F 2.232 =3.93, *p*-value = 0.02) (Figure 1A). In particular, AA individuals perceived βI as significantly less familiar, less intense, and less edible than AG and GG individuals. According to post hoc Tukey HSD tests (α = 0.05), AG and GG carriers did not differ on these four parameters and gave higher scores than AA carriers. Ratings of pleasantness, edibility, familiarity and intensity were not significantly correlated with each other (Pearson R-squared < 0.50).

On a verbal level, the terms most frequently used by AA and GG individuals to describe the smell of βI were general terms such as “medicine”, “perfume”, or “flower”. AG individuals used the word “rose” significantly more than AA individuals (Fisher test two-tailed *p*-value = 0.01) but the difference was not significant compared to the GG group (Fisher test two-tailed *p*-value = 0.28, Figure 1B, Table 1). These results show that OR5A1 genotypes play a significant role in βI perception in Malagasy and confirm the recessive nature of the A allele which is in accordance with studies in other populations [3,18,34].

#### 3.3.2. Detection Scores

The association between the ability to detect βI and rs6591536 genotype was assessed using a detection score test with βI concentrations between 2.05 × 10^−7^ mol/L and 2.05 × 10^−2^ mol/L (*N* = 206) (Figure 2A). Regarding the AG and GG participants, most of them selected the correct vial at 2.05 × 10^−4^ mol/L, and almost all of them smelled a slight or distinct odor at 2.05 × 10^−3^ mol/L (Figure 2A). In contrast, 75% of AA participants indicated the wrong vial at all concentrations, including the highest (2.05 × 10^−2^ mol/L). Because of the 4-alternative forced-choice used in this test, 25% of individuals could have chosen the correct vial at any concentration purely by chance; therefore, it cannot be ruled out that the 25% of AA individuals who chose the correct vial did so solely by chance.

When computing the individual score detection for βI (see Methods), we observed that GG and AG exhibited a significantly lower detection scores than AA individuals (two-tailed Kolmogorov–Smirnov test: AG vs. AA: D = 0.94, *p*-value < 2.2 × 10^−16^, GG vs. AA: D = 0.88 *p*-value = 1.4 × 10^−14^), while there was no significant difference between AG and GG (D = 0.083459, *p*-value = 0.99) confirming again the recessive nature of the allele A.

We then tested, as observed by Jaeger et al., whether the G and A variants of rs6591536 can confer a 100-fold difference in sensitivity in the population. The majority of carriers of the functional form of the olfactory receptor (variant G) are able to detect concentrations of βI at 2.05 × 10^−4^ mol/L; we use this concentration to define normosmic individuals. We therefore consider individuals with a 100-fold detection score, a threshold of 2.05 × 10^−2^ mol/L, as hyposmic. Note that since this limit is the maximum of the concentration tested, individuals classified here as hyposmic may in fact be anosmic, as previously stated, there is a 25% chance of randomly selecting the correct vial (false-positive probability).

Using this definition, forty percent of the Malagasy students appeared to have a reduced sensitivity to βI. A total of 95% of AA carriers were hyposmic while 98.9% of AG and 92.9% of GG were normosmic (Table 2 and Appendix A); thus, being AA homozygote significantly increases the risk of being hyposmic to βI (odds ratio = 704 CI: 150–6793, Fisher Exact Test *p*-value = 2 × 10^−48^; 95%). We also confirm Jaeger et al.’s result showing that the rs6591536 genotype is a reliable predictor of sensory acuity. Indeed, we found that 96.6% of the hyposmic/normosmic phenotype can be predicted by this variant which is strikingly similar to the result obtained by Jaeger et al. (96.3%). These results show that the perceptual ratings and threshold test by genotype distribution in Madagascar highland population is consistent with the populations studied in the Jaeger (2013) study where the G allele dominantly confers the sensitivity to ß-ionone.

When controlling for age and sex difference in βI detection scores, no significant difference has been found (Age: R-squared = 0.01; sex: *p* > 0.9553 Mann–Whitney test).

## 4. Genome Data Analysis

### 4.1. GWAS Analyses

GWAS analyses confirmed that the OR5A1 gene locus on chromosome 11 (Appendix A, Appendix A) is highly associated with B-ionone perception. Specifically, snp rs6591536 is the most strongly associated SNP (*p*-value < 10^−39^). It should be noted that other nearby SNPs including some on olfactory receptors are also associated with B-ionone perception, but more weakly (*p*-value < 10^−7^, Appendix A), this is expected due to their linkage disequilibrium with rs6591536.

### 4.2. Relationship between Insensitivity to βI and Local Ancestry

To assess whether the African genetic introgression influenced the perception of βI in the highland of Madagascar, we evaluated the association between βI detection and the ancestry of the OR5A1 gene by applying the ELAI algorithm on OR5A1 locus (Table 2). On average, 68.7% of the OR5A1 locus analyzed in the population have an East Asian ancestry and 31.3% African ancestry, confirming the minor African ancestry and major East Asian ancestry. We observed a strong influence of ancestry on the OR5A1 locus and the rs6591536 genotype (Table 2 and Appendix A). For instance, the allele frequency of A is 70.4% in individuals with the two OR5A1 loci from East Asian ancestry (EAS-EAS) while the frequency of A is 50% in individuals with two loci from African ancestry (AFR-AFR). We observed that 93.8% of individual carriers of two OR5A1 loci from African ancestry are normosmic while only 51.6% of individual carriers of two OR5A1 loci from Asian ancestry are normosmic (portion in green, Figure 3). Participants with one OR5A1 loci of each origin (AFR-EAS) present an intermediary proportion with 61.9% of participants classified as normosmic to beta-ionone. These results show the strong association between East Asian ancestry and reduced sensitivity to βI (odds ratio = 14 CI: 1.8–110, *p*-value = 0.012; 95%).

## 5. Discussion

In this paper, we aimed to investigate the impact of intercontinental gene flow on βI perception in an admixed population such as the Malagasy population. Our results show that the perception scores and detection test according to the distribution of genotypes in the highland population of Madagascar are consistent with the populations studied in Jaeger’s (2013) study where the G allele dominantly confers sensitivity to ß-ionone. The strong genotype/phenotype association between rs6591536/ßI found by Jaeger et al. (2013) is then confirmed in a Malagasy population. We also showed that in the Malagasy population studied, the frequency of the “insensitive” genotype is lower than that of the “sensitive” genotype due to African gene flow.

The phenotyping of individual perception was performed according to the reliable and rapid olfactory test developed by Rouby et al. in European populations (LCOT: Lyon Clinical Olfactory Test) [27]. This portable kit designed for self-administration and validated for cross-cultural studies was chosen as the most suitable test to effectively assess Malagasy participants’ olfactory abilities in a large-scale setting, compared to a standard staircase methodology. A putative disadvantage of this choice is that the staircase methodology would have been more accurate. However, in the case of βI where we are studying a specific anosmia/hyposmia with at least a factor of 100 difference in terms of threshold level. Therefore, the level of precision is not an issue. This methodological choice was validated by the fact that we found that rs6591536 can predict ßI sensitivity with an accuracy of 96.6%, which is extremely high and very similar to the result obtained by Jaeger et al. (96.3%). Such a high level of prediction, as high as 96.3%, implies that the level of error in phenotyping is necessarily low and therefore that the methodology was adequate.

The extreme similarity between our results and those of Jaeger et al. is surprising, especially considering the extreme differences in genetics, culture, and lifestyle between the study populations, Malagasy and New Zealanders, and Southeast Asian populations. The Malagasy populations are the result of a deep and ancient mixture of Southeast Asian and African island gene flows, which has resulted in uncommon genetic and cultural diversity [35,36]. Multiple genetic or cultural factors could have an impact on the genotype/phenotype correlation. For example, in a previous study, we showed that androstenone-specific anosmia in Madagascar can be significantly influenced by polymorphisms that are less frequent or absent in other populations [36]. In the case of ßI perception, it could have been suggested among various possibilities, that different sets of functional olfactory receptor genes might be expressed in Malagasy. Additionally, these receptors might have been activated by βI, therefore, even individuals carrying insensitive alleles of OR5A1 would be able to detect βI. Consistent with this hypothesis, a series of olfactory receptors (OR5A2, OR4D6, OR4D9, OR2B3, OR52D1, OR1G1) were found to be activated in vitro by βI [14,37,38]. Nevertheless, the extreme similarity between our results and those of Jaeger et al. refutes this assumption. As we confirmed the strong impact of the polymorphism rs6591536 on the perception of βI, it is unlikely that alternative receptors compensate insensitive alleles of OR5A1 in this Malagasy population.

One could argue that cultural factors could modulate the genotype/phenotype relationship. Indeed, it is known that repeated exposure to odors can modify sensitivity to them (as in olfactory training) [39]. The Malagasy students tested in this study are likely to have a very different lifestyle from the populations previously tested [14,18]. Rice is their daily meal and most of them do not have access to western processed foods from supermarkets, which probably makes them less exposed to βI. It could be hypothesized that the level of exposure to βI differs between populations, which could therefore influence the perception of βI. However, this factor does not seem to impact the genotype/phenotype association between rs6591536 polymorphisms and ßI perception.

Many factors can influence the individual phenotype such as sex, genotype, and health status. In the present article, we show that despite all these theoretical possibilities, the studied OR5A1 polymorphism explains 95% of the variability. For example, sex does not influence the threshold of detection of the molecule’s perception. Thus, despite the fact that women are underrepresented in our sample, the computed prevalence of anosmia/hyposmia should be representative of the prevalence encountered in the general population. However, it has to be kept in mind that our study is based on a homogenous sample of young adults (students), and it has been shown that a decrease of the human olfactory perception along with aging [28]. Therefore, in a global population, the strong effect of OR5A1 polymorphism in the perception of beta-ionone might be lessened.

The fact that the Malagasy population is genetically and culturally different from the two populations tested in New Zealand and that it comes from an Asian/African mixture suggests that this association is likely to be found in most populations of the world. The global distribution of the rs6591536 polymorphism can therefore be considered a good proxy for predicting the diversity of ßI perception of current world populations; based on the available genomic dataset, a large variation in the proportion of ßI-insensitive individuals worldwide can be expected [19]. βI is present in many plants, fruits, and by-products (tomatoes, white-fleshed nectarines, violet flowers, grapes, and therefore wine), and is also widely used as a flavor enhancer in the food industry [40,41,42,43]. Therefore, genetic predisposition to βI anosmia/hyposmia could substantially influence the diversity of appraisals and product choice [18,34] and food culture across the world. Future cross-population studies will provide a better understanding of the influence of genetic polymorphisms on dietary practices and culture [22].

Our results demonstrate that it is possible to track the effect of migration and gene exchange on the sensory perception of a population. These results are largely based on the reliability of the estimation of local ancestry. The reliability of the determination of local ancestry depends mainly on the differentiation of the chromosome fragments, which depends on two parameters (1) the age of the mixture and (2) the age of the separation between the two source populations. The older the separation and the more recent the mixing, the more reliable the local ancestry algorithms are. In the case of the Malagasy population, the admixture is relatively recent (1000 years) and the split extremely old (60,000 years). The two source populations living in Southeast Asia and East Africa are therefore extremely differentiated, with almost no gene flow between these two periods. Therefore, algorithms can reliably assign to one of two sources (Asian versus African sources) each locus, in consequence, the results of the different local ancestry estimators are very consistent [44] This suggests that local ancestry might be a tool to follow past the allelic frequency.

Tracking the allelic frequency of rs6591536 in the past population could be an indicator of the sensory phenotype of different migrants who settled in Madagascar and elsewhere. For the present study, populations had sustained contact across the Indian Ocean as early as 4000 BC, which is one of the earliest long-distance maritime contacts [45]. These connections—called proto-globalization—triggered massive transfers of crops, techniques, and ideas across oceans [33,46] and brought Indonesian and African migrants together on the island of Madagascar around 1000 years ago [35]. After the arrival of the migrants, the challenge of colonizing a new territory as well as the extensive exchanges between these two populations caused substantial changes at all levels in their respective ways of life. For settlers on both sides—African and Indonesian—their language, religion and culture, food practices, and technologies evolved and integrated elements of both origins through phenomena such as introgression, mixing, and acculturation [47,48,49]. It has been shown that gene exchanges between populations can impact the phenotypes of the recipient populations [50,51] but very little is known about the influence of gene exchange on their sensory perception, emotions, and culture. Because of the high genetic influence, we suggest that the frequency of ßI-hyposensitivity in the African population that migrated to Madagascar was very limited (18,75%) (Figure 4), in contrast to the frequency of ßI-hyposensitivity in the East Asian ancestry (49%). Analyzed globally, Malagasy participants in this study harbor about 41% of AA “ßI-insensitive genotype” versus 59% of GG and GA “ßI-sensitive genotype” (Figure 4; Appendix A). The SNP rs6591536 has one of the greatest worldwide variations of allele frequency (Figure 4, Appendix A). In sub-Saharan Africa, the prevalence of homozygotes AA associated with hyposensitivity for beta-ionone appears to be consistent and very low. Therefore, the specific Island Southeast Asian origin of the Malagasy ancestor played an important role in the final frequency of beta-ionone anosmia in the Malagasy population. We estimate that the frequency of the insensitive genotype was already around 50% in the Asian source population, which is already lower than in many Asian populations. Since all African populations have a low level of insensitive genotype, it is possible to conclude that in the case of intercontinental contact, mixing with any sub-Saharan population would decrease the percentage of insensitive genotype. These results are consistent with the genetic diversity of the two continents, suggesting that this method seems reliable for estimating the frequency of past phenotypes in the source population. They also show that introgression of African gene flow reduced the percentage of βI hyposmia in the Malagasy highland population. As a proof of concept, we show here evidence of the impact of gene flow from African populations on the olfactory system of highland settlers. Our work confirms the two preconditions announced in the introduction. Thus, by studying different populations with different admixture ratios and exploring the effect of ßI perception on the cultural aspect such as food choice or scent, we should be able to investigate the effect of gene flow on the cultural aspect.

## 6. Conclusions

In conclusion, the use of bioinformatics tools such as local ancestry [59] or ADMIXTURE [60] allows the calculation of allelic frequencies in ancient populations, which opens up possibilities to discuss the history of olfactory perception and emotions (disgust and pleasure) and to speculate for example on their impact on the feeding behavior of past populations. As suggested by Jaeger et al., this could also apply to very ancient populations such as Neanderthals whose perceptual abilities can be inferred on the basis of polymorphisms in their OR genes. The case of ßI perception is particularly noteworthy because of the high prevalence of βI in many products and could lead to various dietary behaviors and practices. Further studies are needed to test this hypothesis, but this approach could lead to interesting findings regarding the co-diffusion of genes and cultural practices across the globe.

## Figures and Tables

**Figure 1 brainsci-11-01405-f001:**
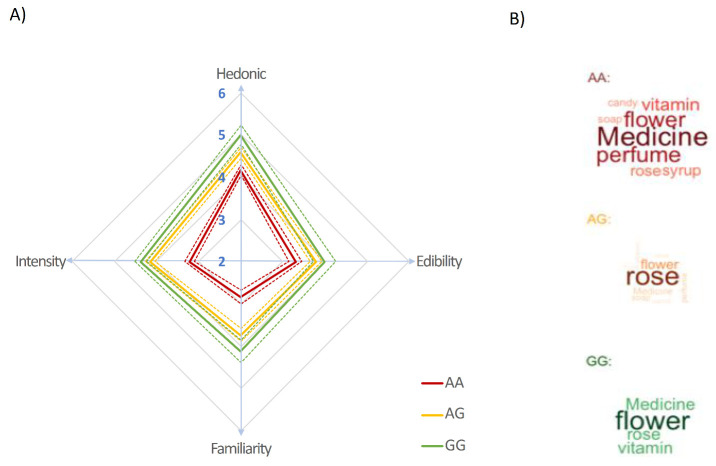
Perceptual evaluations of βI at a supraliminal concentration by Malagasy participants according to their genotype. All participants meeting the inclusion criteria are shown here (*n* = 235). (**A**) Average ratings on the perceptual scales (Hedonic, Edibility, Familiarity, and Intensity), mean (solid lines) standard error of the mean (dashed lines). (**B**) Word clouds of descriptors used by the participants according to their genotype (AA, AG, and GG). The size of each descriptor is proportional to the frequency of its usage by the participants.

**Figure 2 brainsci-11-01405-f002:**
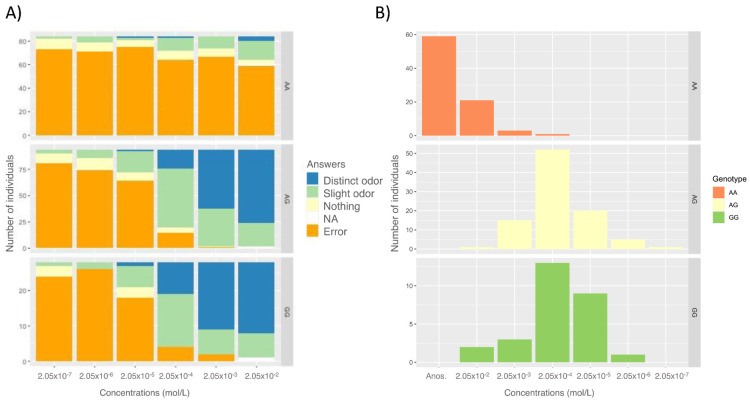
Detection of beta-ionone according to odor concentration and genotype. (From the 235 initial participants only the 206 who have performed the threshold detection task are represented here.) (**A)** Percentage of participants choosing the wrong vial (Error) and evaluating the vial they selected in the detection threshold test as smelling nothing, a slight odor, or a distinct odor, for each concentration (mol/L) and according to their genotype. NA represents the proportion of participants who selected the correct vial but did not rate it. (**B**) Computed detection threshold for beta-ionone according to genotype. (Anos = anosmic to beta-ionone).

**Figure 3 brainsci-11-01405-f003:**
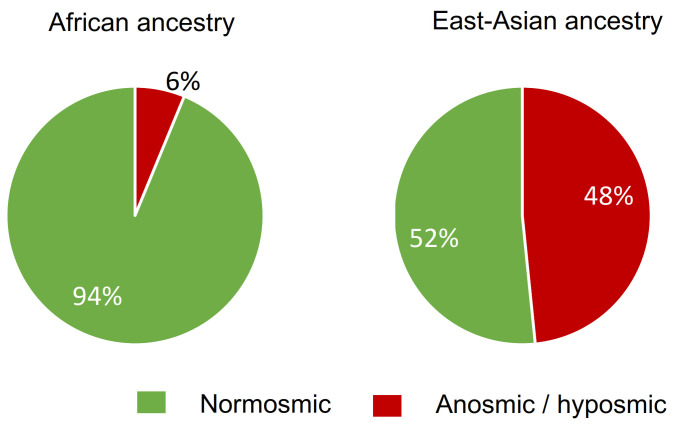
Distribution of participants’ beta-ionone sensitivity to the ancestry of their two OR5A1 loci. Only individuals with homogenous ancestry are represented, Individuals with dual ancestry (one OR5A1 locus from each ancestry) are excluded from this representation.

**Figure 4 brainsci-11-01405-f004:**
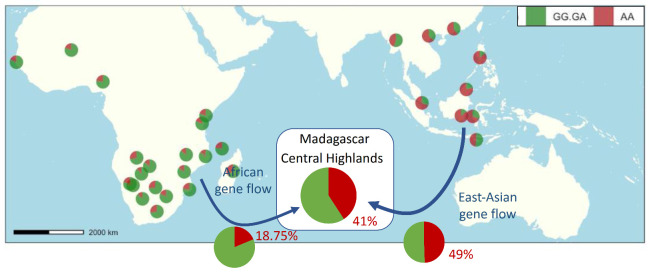
Distribution of “normosmic” (GG and GA) and “hyposmic” (AA) genotypes of r6591536 across populations from the western and eastern Indian ocean. Pie charts represent the proportion of genotypes across populations and are positioned according to the location of sampling [35,44,52,53,54,55,56,57,58]. The proportion of insensitive genotypes appears in red (Malagasy population and gene flows). The proportions in the gene flows are computed accordingly to the genotype frequency of Malagasy participants with the same origin for both chromosomes.

**Table 1 brainsci-11-01405-t001:** Qualitative description of beta-ionone: number of participants (per genotype and total) using each descriptor (only descriptors used by more than five participants are shown).

Words	AA	AG	GG	Total
rose	6	19	3	28
flower	9	9	5	23
medicine	11	6	3	20
perfume	9	5	2	16
vitamin	7	4	3	14
soap	4	5	2	11
candy	4	3		7
syrup	6	1		7
fruits	2	3		5

**Table 2 brainsci-11-01405-t002:** Ability to detect beta-ionone according to (i) the participants’ genotype (column); (ii) the chromosome origins (column) (AFR-AFR: two loci with African local ancestry; AFR-EAS: one locus with African local ancestry and one with East Asian ancestry; EAS-EAS: two loci with East Asian local ancestry) (from the 235 initial participants only the 206 who have performed the threshold detection task are represented here).

		Genotype Frequency			
		AA	GA	GG	Total
Locus ancestry	**AF/AF**	18.75% (3)	62.5% (10)	18.75% (3)	16
	**AS/AF**	36.08% (35)	46.39% (45)	17.53% (17)	97
	**AS/AS**	48.94% (46)	42.55% (40)	8.51% (8)	93
	Total	40.58% (84)	45.89% (94)	13.53% (28)	206

## Data Availability

The data presented in this study are openly available in Appendix A.

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
