# Peer review of "African Gene Flow Reduces Beta-Ionone Anosmia/Hyposmia Prevalence in Admixed Malagasy Populations"

_brainsci, 2021, doi:10.3390/brainsci11111405_

Round 1
Reviewer 1 Report
I think that the revised version of the manuscript is improved.
Author Response
Thank you for your comments
Reviewer 2 Report
Manuscript ID: brainsci-1425736
African gene flow reduces beta-ionone anosmia/hyposmia prevalence in admixed Malagasy populations
RECOMMENDATION – MINOR REVISION
The paper Razafindrazaka et al. is very interesting, pointing to genetic background of odour perception in a Malagasy population. In this study, the Authors aimed to investigate the impact of intercontinental gene flow on β-Ionone (βI) perception in an admixed population such as the Malagasy population. The results show that the perception scores and detection test according to the distribution of genotypes in the highland population of Madagascar. The results are consistent with the populations studied in Jaeger's (2013) study where the G allele dominantly confers sensitivity to ß-ionone. The strong genotype/phenotype association between rs6591536/ßI found by Jaeger et al. (2013) is now confirmed in a Malagasy population. The Authors also showed that in the Malagasy population, the frequency of the "insensitive" genotype is lower than that of the "sensitive" genotype due to African gene flow.
Comments and Suggestions:
- Since men and women took part in the study, it might be interesting to check the influence of gender on the parameters tested.
- Fig. 2 – is too small, especially part A. It is unreadable.
- Fig. 3 – figure legend covered the scheme.
- Please specify, why in each figure/analyzes different number of people took part in the research.
- Twice as many men as women participated in the study, it would be interesting to discuss this fact.
Author Response
Please see the attachment

This manuscript is a resubmission of an earlier submission. The following is a list of the peer review reports and author responses from that submission.
Round 1
Reviewer 1 Report
The project is highly interesting however two main gaps I should noted:
A- what is the olfactory phenotype of the participants? (to better understand the question see Mazzatenta et al 2016 Oncotarget, which is lack in the References too) could be an important variation across subjects
B- what is the 'general' olfactory threshold of partecipants? (tested with Cain or Olfactory Smart Threshold test? see in Frontiers in Medicine 2020 for example)
without addressing these points, for example, table 2 have no sense to me. The authors think that the olfactory phenotype is juvenile and olfactory threshold is normosmia ... but should demostrate ... for example, in the exclusion criteria should be nasal polyposis or rhynitis after citological examination.
major points:
- line 52. I would like to suggest to the authors that they explain whether androstenone and B-ionone are to be regarded in our species only as odorants or also as pheromones, and then explain the difference between odorants and pheromones.
- I would suggest a more in-depth statistical analysis for "psychophysical" tests
- it is possible to better discriminate your results as in the following scale: normosmic, hyposmic, severe hyposmic and anosmic?
minor points:
- please remove instruction for authors line 38-46!
- line 54 check 'anomic'? maybe is anosmic
- line 84 explain what is rs6591536
- check some other imperfection in the paper
- check the References
Author Response
"Please see the attachment."

Reviewer 2 Report
In their manuscript, Razafindrazaka et al. present their work showing that African gene flow in Madagascar increases the beta-ionone normosmia of the population. To do so, they selected participants who they genotyped to determine their ancestry (African or East Asian) as well as the genotype of OR5A1, a receptor already shown to be strongly involved in the beta-ionone perception. They also monitored the perception and detection threshold of the participant to beta-ionone using QDA, free description, and six 4-alternative forced-choice tests. Doing so, they were able to link the ancestry to the OR5A1 genotype and the beta-ionone perception, showing that African ancestry contributes to a better perception of this molecule. The shown results are extremely convincing, and the interest of the presented study is very high to understand how ancestry, genotype, culture, and population flow influence odor perception and so consumer choices. The reviewer is convinced that the presented manuscript should be published following the suggested modifications.
Concerning the supporting information, the manuscript lacks side analysis that would support the data from the main manuscript. For example, it would be very informative to show the plot of the associated region on chromosome 11 and describe at least the prevalence or not of SNP on the other ORs known to respond to beta-ionone. The authors could also analyze and discuss the variability in genotype in both “flow” (African and East Asian) as it seems the specific origin could seriously influence the genotype of the final population. The authors also need to provide all their raw data in the supplementary information (for each participant: perception description, threshold, genotype, ancestry, etc).
Concerning the method, it seems the author omitted to include a negative control odor or a precheck of the capacity of the participant to smell, which would show that the hyposmic participants have specific hyposmia to beta-ionone. Could the authors explain their choice?
The author should also add a critical discussion on the level of confidence of the ancestry determination.
Table 2 is very rich in information. It would help the reading to separate these data. The genotype frequency is what matters when we want to show introgression so a clear table showing only these data is necessary.
Last, Figure 4 seems of high interest but it is hard to understand which data are shown here. Please describe how you get the genotype data of the different populations, what are these two pie charts under the arrows, and how is obtained the boxed pie chart. Also, it is necessary to discuss the prevalence of both ancestries in the Malagasy central highlands population to explain the results shown here, as well as a discussion on the prevalence of one of the East Asian populations as the genotype seems to change a lot in this region.
Minor changes:
L54 anosmic
L185 Af[24]rican
L338 to 342 this sentence needs to be cut.
Author Response
"Please see the attachment."

Reviewer 3 Report
I find that the study submitted by RAZAFINDRAZAKA et al. is really interesting and extremely topical in the field of research of the olfactory system and on the factors that determine its individual variability. Precisely for this reason, I think that the authors should cite in the introduction recent findings on the role that the rs2590498 polymorphism of OBPs plays on the greater or lesser threshold of olfactory perception presented by individuals.
Author Response
"Please see the attachment."
